# Challenging Cases of Aortic Prosthesis Dysfunction, the Importance of Multimodality Imaging, a Case Series

**DOI:** 10.3390/diagnostics11122305

**Published:** 2021-12-08

**Authors:** Valeria Pergola, Giulio Cabrelle, Giorgio De Conti, Giulio Barbiero, Donato Mele, Raffaella Motta

**Affiliations:** 1Department of Cardio-Thoraco-Vascular Sciences and Public Health, University of Padua, 35128 Padua, Italy; valeria.pergola@aopd.veneto.it (V.P.); donato.mele@unipd.it (D.M.); 2Department of Medicine—DIMED, University of Padua, 35128 Padua, Italy; raffaella.motta@unipd.it; 3Radiology Unit, Padova University Hospital, 35128 Padua, Italy; giorgio.deconti@aopd.veneto.it (G.D.C.); giulio.barbiero@aopd.veneto.it (G.B.)

**Keywords:** ECG-gated multidetector computed tomography (MDCT), echocardiography, prosthetic heart valve (PHV), valvular dysfunction

## Abstract

ECG-gated multidetector computed tomography (MDCT) is a promising complementary technique for evaluation of cardiac native and prosthetic structures. MDCT is able to provide a broader coverage with faster scan acquisition times that yield higher spatial and temporal resolution for cardiac structures whose quality may be affected by artifacts on ultrasound. We report a case series about the most challenging complications occurring after prosthetic aortic valve implantation in four patients: pannus, paravalvular leak, prosthesis’ misfolding and subaortic membrane reformation. In all the cases, enhanced MDCT using a retrospective protocol provided accurate 3D morphoanatomic information about cardiac and extracardiac structures, improving and speeding up the correct diagnosis and treatment planning. Integrated imaging, in particular with MDCT, is now the present, and it will increasingly be the future in the assessment of cardiac structural pathology.

## 1. Introduction

Transthoracic (TTE) and transesophageal echocardiography (TEE) represent the primary noninvasive tests in the follow-up of patients with prosthetic valve diseases. Cardiac computed tomography (CCT) is generally employed when other imaging modalities fail to reach a definite diagnosis of valve dysfunction [1].

ECG-gated multidetector CT (MDCT) is becoming a useful technique for the evaluation of cardiac anatomy, with an increasingly solid evidence-based role [2]. Multidetector and dual-source scanners are able to provide a broader coverage with faster scan acquisition times that yield higher spatial and temporal resolution for cardiac structures.

In addition to its powerful diagnostic accuracy in coronary artery disease, a growing indication of MDCT is the assessment of native and prosthetic heart valves (PHVs). In the latter case, several artifacts, such as mechanical acoustic shadowing, reverberations, refraction and mirroring, may develop, affecting the imaging quality on ultrasound (US) [2,3,4]. In patients with PHVs, dynamic 4D imaging provided by MDCT can provide detailed information about the morphology and mobility of the PHV’s elements, detecting the cause of valve dysfunction (thrombus, pannus, calcific degeneration of PHV leaflets) and quantifying its severity [2,3,4,5,6,7,8], despite the fact that image quality may be affected by beam hardening and cardiac motion artifacts.

We report a case series about the role of MDCT in PHV dysfunction importance of multimodality imaging (echocardiography integrated with MDCT) to ensure correct diagnosis and treatment.

## 2. Case Series

CT was performed with axial imaging using a third-generation 320 × 0.5-mm detector row CT unit (Aquilion ONE ViSION Edition; Toshiba Medical Systems, Otawara, Japan). Gantry rotation time: 350 ms; automatic exposure control (SURE bExposure 3D, Toshiba Medical Systems, SD 110 for contrast-enhanced images); 512 × 512 matrix; retrospective protocol; section thickness of 0.5 mm with 0.25 mm increments using kernel FC03; iterative reconstruction AIDR3D standard (Toshiba Medical Systems); intravenous contrast: 50–80 mL Iomeron^®^ 400 mg/mL (Bracco Imaging Italy s.r.l., Milan, Italy) with 5 mL/s flow; heart rate was set between 50 and 60 beats per minute with intravenous Metoprolol. The data were transferred to an external workstation (Vitrea2 FX version 6.3, Vital Images, Plymouth, MN, USA) providing multiplanar reformation (MPR), volume rendering technique (VRT) and cineview reconstructions.

US scans were performed with Philips Epiq Cvx (Philips, Andover, MA 01810, USA) with sectorial probes: 3.5 MHz (TTE) and 3–8 MHz (TEE).

### 2.1. Case 1

Female, 83 years old (y/o), with hypertension, hypercholesterolemia, carotid vasculopathy (type III, AHA) and history of (h/o) smoking affected by degenerative aortic stenosis, underwent valvular replacement with a St. Jude 21 mm mechanical prosthesis in 2000. Twenty years later, she was hospitalized for respiratory distress. TTE showed left ventricle (LV) dysfunction with severe prosthetic valve stenosis (aortic acceleration time (AAT): 140 ms, transaortic maximum speed: 4.8 m/s, maximum/median gradient: 90/52 mm Hg, indexed effective orifice area (EOA): 0.3 cm^2^/mq, EF: 35%). TEE showed hypomobility of the anterior leaflet. Due to the shielding from the prosthesis, it was unclear if there was a thrombus or a pannus (Appendix A). As it is possible to differentiate between a pannus and a thrombus due to their different radiological density (HU > 145 and > 90, respectively) [9], MDCT was performed, and it showed that the anterior aortic leaflet was stuck and surrounded by hypodense tissue (Hounsfield units (HU): 203.8) interposed between native and prosthetic annuli (effective orifice area (EOA): 45 mm^2^, EOA/0.15) indicating a pannus (Figure 1a–c). This information was of utmost importance as instead of staring anticoagulant treatment, the patient directly underwent repeat surgical repair with a bioprosthesis. The diagnosis of pannus was confirmed by pathology.

### 2.2. Case 2

Female, 44 y/o, affected by mitral valve (MV) dysplasia (parachute valve with double medioposterior papillary muscle) and subaortic stenosis caused by a fibromuscular ring, underwent subaortic membrane resection and septal myectomy in 1989. Due to worsening exertional dyspnea and persistence of subaortic stenosis, a St. Jude Regent 17 mm was implanted in 2006 (40 y/o) with improvement of her physical condition. In the last 2 years, TTE detected a progressive increase of the intraventricular gradient with LV hypertrophy (maximum speed, 4.1 m/s, maximum/median gradient: 64/39 mm Hg). TEE performed in May 2020 showed normal excursion of the prosthesis’ leaflets and confirmed severe subaortic stenosis (speed: 5.5 m/s, maximum/median gradient: 120/63 mm Hg) (Appendix A). New subaortic membrane formation (SAM) was suspected but not clearly detected by TEE. MDCT provided accurate 3D reconstructions of the LV outlet tract (LVOT) with a better topographic assessment of the new SAM and its surrounding structures. The SAM was located 7 mm below the aortic prosthetic annulus, with the maximum thickness of 5 mm and hemicircumferential extension along the interventricular septal surface. This information was crucial to guide surgical excision of the SAM (Figure 2a,b).

### 2.3. Case 3

Male, 80 y/o, with a metabolic syndrome. He underwent thromboendarterectomy because of right internal carotid artery serrate stenosis. Due to bivasal critical coronary stenosis (anterior descending (DA) and left circumflex (LCx)) and severe degenerative aortic stenosis, he underwent coronary artery bypass graft (CABG: left internal mammary artery (LIMA-IVA)) and aortic bioprosthesis implantation (Intuity 25 mm) in 2019. Ten months after surgery, he started developing intermittent fever with serial hemocultures growing *Enterococcus faecalis*. TTE detected paravalvular regurgitation (PVR) with focal hyperechogenic thickening of the leaflets. Diagnosis of endocarditis was made, and antibiotic treatment was started (meropenem shifted to ampicillin and ceftriaxone according to the antibiogram). TEE showed a pulsatile perivalvular pseudoaneurysm in the mitroaortic intervalvular fibrosa (Appendix A). MDCT was performed a few hours later, confirming the presence of a pseudoaneurysm with the maximum axial size of 15 × 10 × 30 mm communicating with LVOT through a 5 mm window, and also detected a periaortic abscess in the anterolateral side of the vessel with longitudinal extension of 4 cm, which was only poorly detected by TEE (Figure 3a–c).

### 2.4. Case 4

Male, 69 y/o, with hypertension, hypercholesterolemia and previous myocardial infarction. He was affected by severe degenerative aortic stenosis and underwent trans-catheter aortic valve replacement (TAVR) with LOTUS Edge 27 mm in April 2020. TTE performed a few days after the TAV implantation detected an increased transprosthesis gradient (maximum/median gradient, 78/52 mm Hg) in the absence of fever or positive hemoculture. TEE showed hypomobility of the noncoronary cusp of the bioprosthesis (Appendix A). Valve’s thrombosis was suspected and heparin administration was started. MDCT detected a paravalvular leak caused by misfolding of the prosthesis’ frame; the suspicion of valve thrombosis was also confirmed by the finding of two hypodense appositions at the lower edge of the valve. The patient underwent balloon valvuloplasty with complete resolution of the valvular dysfunction (Figure 4a–c).

## 3. Discussion

The most frequent nonstructural complications occurring after the implantation of PHVs include thrombus (0.3–8%) or pannus (0.2–4.5%), paravalvular leak (2–10%), endocarditis (1–6%) with abscess or pseudoaneurysm formation and aortic dissection (0.6%) [3,10,11,12,13]. In our case series, MDCT was able to better define PHV morphology and function. Moreover, it provided a detailed 3D overview of the whole heart anatomy.

The distinction between a pannus and a thrombus may be challenging [3,12]. It has been shown by recent studies that MDCT is a very sensitive technique capable of differentiating a pannus from a thrombus by detecting its exact location and measuring its attenuation [14,15]. A pannus usually affects the ventricular side of the PHV and has attenuation similar to myocardium [12]; on the contrary, a valvular thrombus usually has lower attenuation and preferentially involves the aortic side of the prosthesis [12]. In case 1, attenuation of the hypodense perivalvular tissue was 50 Hounsfield units (HU), the LV wall [16]. Moreover, with the high spatial resolution of MDCT, it is possible to perform accurate calculations of the leaflet’s opening angles (Figure 1c), the maximum pannus width and the pannus encroachment ratio, all of which are associated with the magnitude of PHV dysfunction [17].

In case 2, MDCT confirmed the presence of a subaortic membrane (SAM), which is one of the most frequent causes of subaortic stenosis. MDCT, due its better accuracy compared to US [18], was able to provide accurate 3D morphoanatomic information about the size, locations and extent of the fibromuscular bulge, the knowledge about which is a fundamental requirement to plan a surgical intervention [19], especially in this complicated patient affected by Shone complex.

In case 3, TTE detected paravalvular regurgitation (PVR), which raised the suspicion of endocarditis [2,11]. TEE detected a paravalvular pseudoaneurysm but failed to demonstrate the anterior abscess, and the overall quality of images was worse compared to MDCT due to acoustic shadowing. MDCT was indeed useful in confirming the presence and the size of the pseudoaneurysm by analyzing the mitroaortic intervalvular region with MPR [20,21].

In case 4, the 3D anatomical model provided by MDCT clarified the stent’s frame’s crumpling morphology, guiding the choice of the right tools for angioplasty. MDCT integrated with metal artifact reduction filters plays an important role in determining the morphology and location of the frame’s misfolding in percutaneous valves [20,22,23], caused by native aortic valve/annulus calcifications, valve undersizing or mal-positioning [2].

Indeed, the presence of PHV-related artifacts could reduce the image quality in MDCT as well, but recent studies agree on the incremental value over TEE in the assessment of anatomical structures near the prosthesis, like periaortic fat tissue, and in the detection of related pathology, such as pseudoaneurysms [20].

In Table 1, we summarized the advantages and disadvantages of the two techniques, stressing the importance of using all the available tools to achieve the best management and treatment for any single patient. The spread of MDCT is only limited by radio exposition and intravenous contrast use (Appendix A), but with the last-generation machines, the radiation dose estimation for a retrospective study can be around 4 mSv [24], giving the possibility to provide detailed overviews of cardiac and extracardiac structures.

## Figures and Tables

**Figure 1 diagnostics-11-02305-f001:**
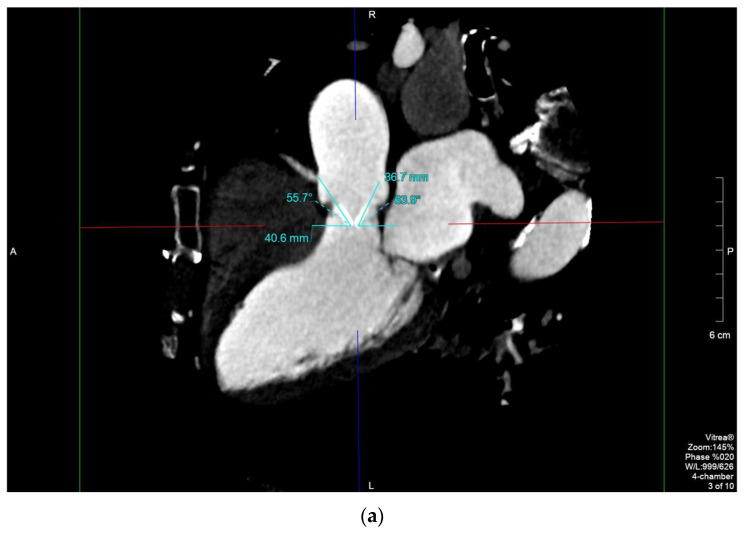
MPR of a St. Jude PHV (10% of the RR interval): long-axis view showing the maximum opening angle of the anterior and posterior leaflets, 55.7° and 63.9° (normal range, 75–90°), respectively (**a**); region of interest (ROI) on the periprosthetic hypodense tissue (**b**); 3D anatomical model of the PHV involved by the pannus (**c**).

**Figure 2 diagnostics-11-02305-f002:**
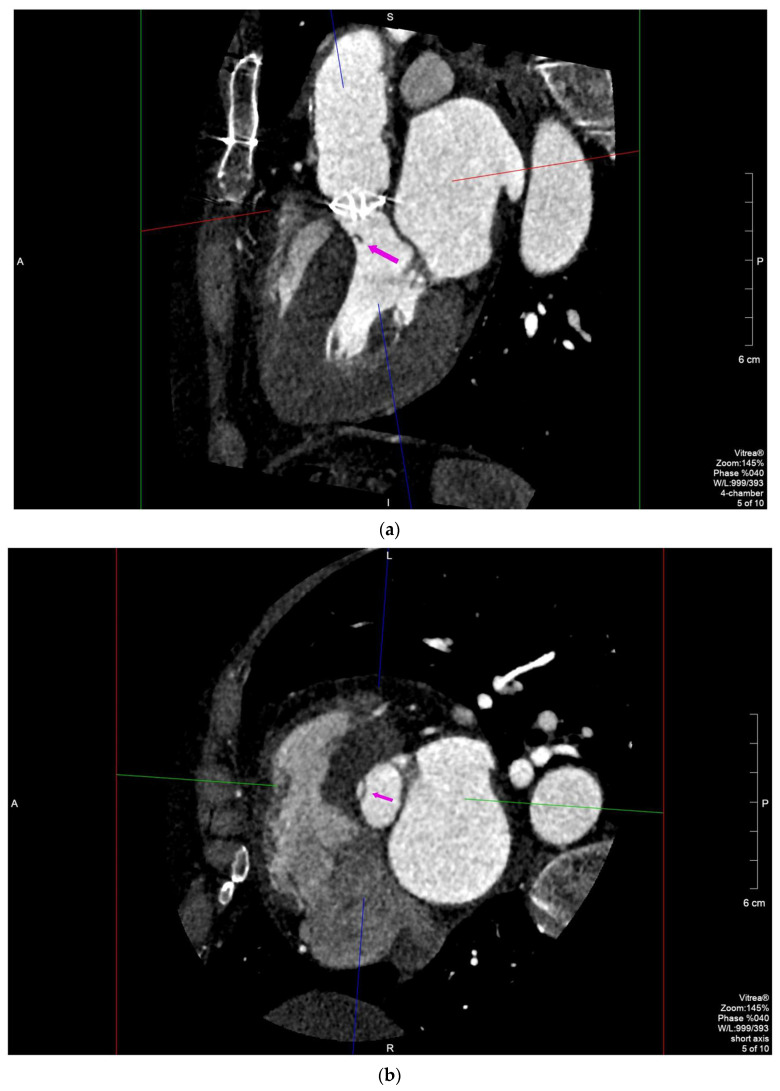
Long-axis (**a**) and short-axis (**b**) MPR focusing on the subaortic membrane (arrow) located 7 mm below the aortic annulus, with the maximum thickness of 5 mm and hemicircumferential extension along the interventricular septal surface.

**Figure 3 diagnostics-11-02305-f003:**
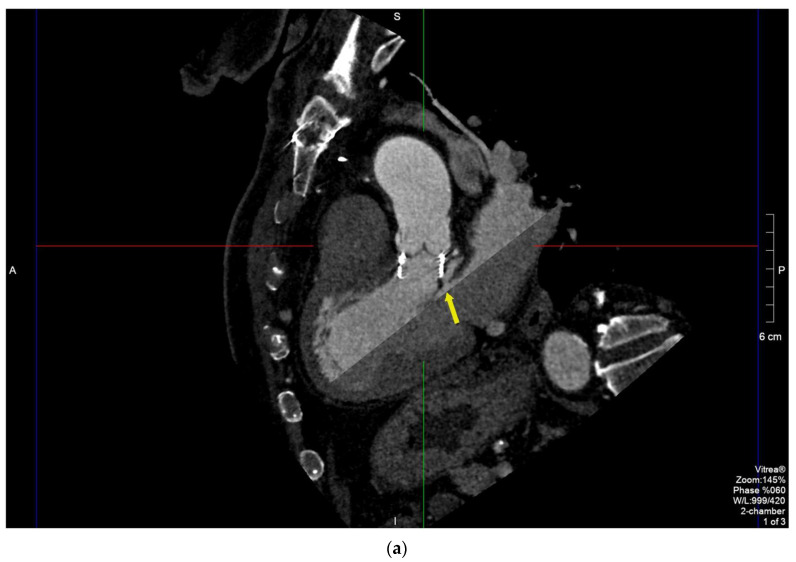
MPR of the aortic annulus to better analyze the pseudoaneurysm (arrows) and its relations with contiguous structures. Long-axis (**a**,**b**) and short-axis views (**c**) of the peri-prosthesis abscess with discontinuous longitudinal extension for 4 cm communicating with LVOT through a 5 mm window.

**Figure 4 diagnostics-11-02305-f004:**
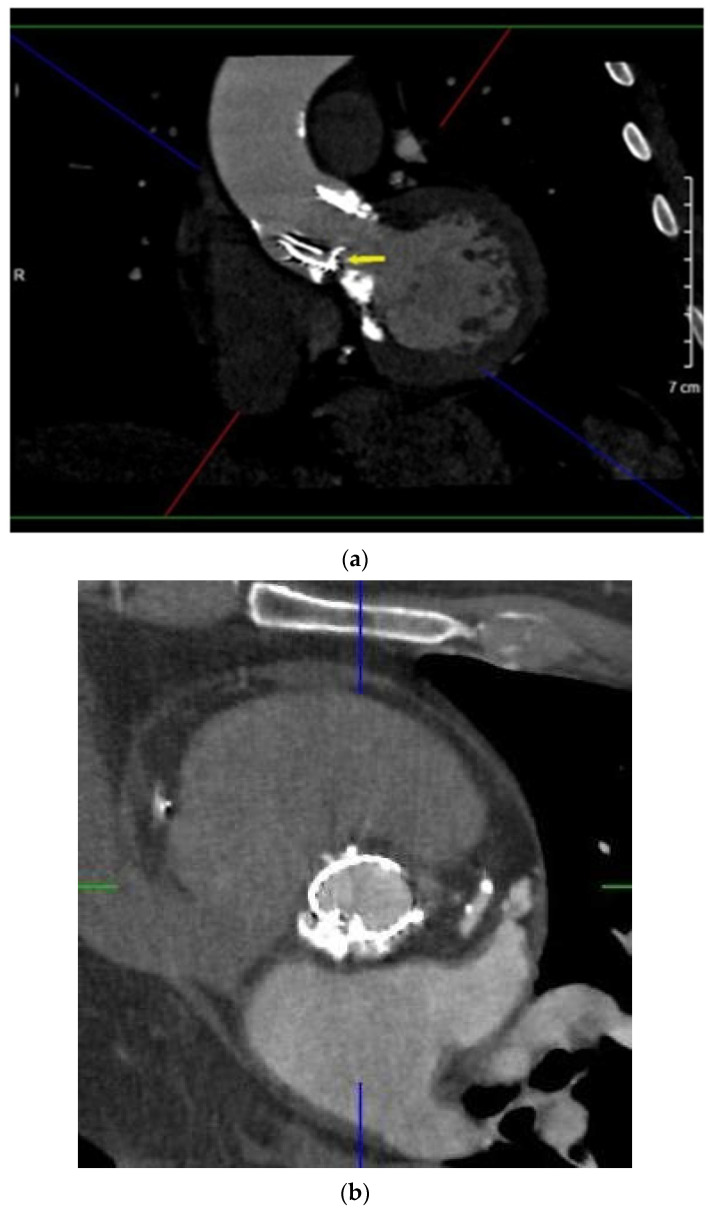
Oblique MPR (**a**), axial (**b**) and 3D anatomical model (**c**) of transcatheter PHV (LOTUS Edge 27 mm) focusing on its frame’s partial crumpling (arrow). A cluster of calcifications can be noted under the PHV.

**Table 1 diagnostics-11-02305-t001:** Comparison between the TEE and MDCT techniques (advantages and disadvantages).

2D and 3D TEE	Cardiotriggered MDCT
Worse spatial resolution	Better spatial resolution
Better temporal resolution	Worse temporal resolution
Harmful for patients (esophageal tear, bleeding)	Harmful for patients (radio exposition, contrast nephropathy, allergic reactions)
Able to measure gradients	Unable to measure gradients
Unbale to evaluate extracardiac structures	Able to evaluate extracardiac structures (large FOV)
Bed side technique	Radiology department facility
Operator-dependent technique	More accurate measurement [15]
Invasive technique	Noninvasive technique

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
