# Peer review of "Challenging Cases of Aortic Prosthesis Dysfunction, the Importance of Multimodality Imaging, a Case Series"

_diagnostics, 2021, doi:10.3390/diagnostics11122305_

Round 1

Reviewer 1 Report

The case series presents several cases of PHV malfunction when MDCT had greatly influenced the diagnostic process. The presented cases are interesting and educational for the readers who treat patients with PHVs.

A few remarks:

General remark:

The authors themselves present extensive literature regarding MDCT use for detecting PHV malfunctions, thus making the innovative value of this manuscript questionable. Although this is not an illative study but a descriptive, still one may suggest adding a paragraph describing what does this study adds to current knowledge.

Introduction:

The shortcomings of echocardiography in assessment of PHT function are elaborated clearly, yet some problem with image quality occurs when using MDCT and these should be presented as well.

Case 1:

Please elaborate how the MDCT findings differentiate between a thrombus and pannus (briefly and specifically for this case as this is detailed in general in the discussion), and how this differentiation changed the treatment strategy (e.g., w/o MDCT results the patients would have been started on a heparin/thrombolysis which would not be beneficial and possibly dangerous).

Case 2: As this patient would have probably been referred to surgery for treatment of the increased intra-cardiac pressures in which the surgeon would have visualized the membrane please emphasize what is added benefit for this patient with the prior knowledge about the sub-aortic member (i.e. how did this knowledge change the surgical approach for the benefit of the patient).

Case 3:

“He underwent thrombo-endarterectomy in 2018” In which artery and due to what clinical problem.

Also, again, describe how did the finding in the CT aid the treating physician in the treatment of the patient as the finding was already detected by the TEE.

Discussion:

“The spread of MDCT is only limited by radio-expo-sition (Table-2), but with the last generation machines, the radiations dose estimation for a retrospective study can be around 4 mSv [24], giving the possibility to provide detailed overviews of cardiac and extracardiac structures.”

As most protocols of MDCT require IV contrast use this is another significant limitation of this modality.

Table 1:

Left side “Not harmful for patients” is not accurate. Some patients (invariably the most complicated) are unable to undergo the sedation required for TEE. Also, there is validated data about complication during TEE (i.e. oesophageal tear, bleeding etc.) that must be weighted.

On the other side the use of IV contrast must be mentioned - as complications of contract nephropathy and allergic reactions must be weighted.

Table 2: One feels that this table is not very informative and, if authors prefer, be but in a supplementary document and not in the main article.

Author Response

Dear Reviewer, 

first of all, we would like to thank you for Your time and suggestions to improve the quality of our manuscript. We have answered all Your questions and we have strictly followed Your advice.

Particularly: 

  • Introduction: The shortcomings of echocardiography in the assessment of PHT function are elaborated clearly, yet some problem with image quality occurs when using MDCT and these should be presented as well. It was specified in the conclusions. 
  • Case 1: Please elaborate how the MDCT findings differentiate between a thrombus and pannus (briefly and specifically for this case as this is detailed in general in the discussion), and how this differentiation changed the treatment strategy (e.g., w/o MDCT results the patients would have been started on a heparin/thrombolysis which would not be beneficial and possibly dangerous). We edited the text according to the literature.
  • Case 2: As this patient would have probably been referred to surgery for treatment of the increased intra-cardiac pressures in which the surgeon would have visualized the membrane please emphasize what is added benefit for this patient with the prior knowledge about the sub-aortic member (i.e. how did this knowledge change the surgical approach for the benefit of the patient). We edited the text according to Your request.
  • Case 3: “He underwent thrombo-endarterectomy in 2018” In which artery and due to what clinical problem. It was specified in the text. Also, again, describe how did the finding in the CT aided the treating physician in the treatment of the patient as the finding was already detected by the TEE. The text was edited.
  • Discussion:“The spread of MDCT is only limited by radio-exposition (Table-2), but with the last generation machines, the radiations dose estimation for a retrospective study can be around 4 mSv [24], giving the possibility to provide detailed overviews of cardiac and extracardiac structures." As most protocols of MDCT require IV contrast use this is another significant limitation of this modality. We added it in the text.
  • Table 1: Left side “Not harmful for patients” is not accurate. Some patients (invariably the most complicated) are unable to undergo the sedation required for TEE. Also, there is validated data about complications during TEE (i.e. oesophageal tear, bleeding etc.) that must be weighted. On the other side the use of IV contrast must be mentioned - as complications of contrast nephropathy and allergic reactions must be weighted. The table was edited.
  • Table 2: One feels that this table is not very informative and, if authors prefer, be but in a supplementary document and not in the main article. It was moved to the supplementary materials.

Reviewer 2 Report

With great interest we read the case series of MDCT in patients with valvular disease of Dr Pergola and coworkers. 

My comments are as follows

  1. Were there any patients with atrial fibrillation. How did the authors manage bradycardia in those patients?
  2. Regarding the CT methodology, the acquisition included the whole RR duration? Were there reconstructions in different phases of the cardiac cycle? How did this help in the different cases?
  3. IF the acquisition was performed in the whole RR, how is it possible the DLPs to be so low in cases 1 and 2? Please provide also acquisition parameters and BMIs for every case if possible
  4. Case 1: Could a figure of the pannus measured with HU be provided? Which cutoff for differential diagnosis between pannus and thrombus was used?
  5. Case 2: please provide an explanation of the abbreviation S/P
  6. Case 3: Instead of bi-vasal the term two-vessel could be used. Instead of DA use the whole term diagonal branch
  7. Case 4: Please provide an axial image where the area of malexpansion would be clearer
  8. From figures 1 and 2 please remove the word paraseptal

Author Response

Dear Reviewer,

first of all, we would like to thank you for Your time and suggestions to improve the quality of our manuscript. We have answered all Your questions and we have strictly followed Your advice. Particularly: 

  1. Were there any patients with atrial fibrillation? No, they were all sinus rhythm. How did the authors manage bradycardia in those patients? Bradycardia was managed by intravenous metoprolol and it was specified in the text.
  2. Regarding the CT methodology, the acquisition included the whole RR duration? Were there reconstructions in different phases of the cardiac cycle? How did this help in the different cases? All MDCT were performed with retrospective protocol and reconstructions were performed from 0% to 90% of RR, every 10%. 
  3. If the acquisition was performed in the whole RR, how is it possible the DLPs to be so low in cases 1 and 2? Please provide also acquisition parameters and BMIs for every case if possible. In case 1 and 2 only 80 kw were delivered. The table was edited according to Your requests and DLP were recalculated.
  4. Case 1: Could a figure of the pannus measured with HU be provided? We provided it. Which cutoff for differential diagnosis between pannus and thrombus was used? 145 HU and it was specified in the text according to literature.
  5. Case 2: please provide an explanation of the abbreviation S/P. The text was edited.
  6. Case 3: Instead of bi-vasal the term two-vessel could be used. Instead of DA use the whole term diagonal branch. Done.
  7. Case 4: Please provide an axial image where the area of malexpansion would be clearer. We provided it.
  8. From figures 1 and 2 please remove the word paraseptal. It was removed.

Round 2

Reviewer 2 Report

The authors have adequately addressed all my comments. The manuscript has been improved substantially